# *Phellinus baumii* Polyphenol: A Potential Therapeutic Candidate against Lung Cancer Cells

**DOI:** 10.3390/ijms232416141

**Published:** 2022-12-17

**Authors:** Xue Liu, Shiyao Cui, Caiyun Dan, Wenle Li, Hongqing Xie, Conghui Li, Liangen Shi

**Affiliations:** 1College of Animal Sciences, Zijingang Campus, Zhejiang University, Hangzhou 310058, China; 2College of Life Sciences, Yungu Campus, Westlake University, Hangzhou 310058, China

**Keywords:** antitumor activity, caspase-dependent intrinsic mitochondria apoptosis, cell cycle arrest, network pharmacology, *Phellinus baumii* polyphenols, UPLC–ESI–QTOF–MS

## Abstract

*Phellinus baumii*, a fungus that grows on mulberry trees and is used in traditional Chinese medicine, exerts therapeutic effects against various diseases, including cancer. Polyphenols, generally considered to be antioxidants, have antitumor and proapoptotic effects. In this study, we identified the composition of *Phellinus baumii* polyphenol (PBP) and characterized its 17 chemical components by UPLC–ESI–QTOF–MS. Furthermore, to clarify the potential mechanism of PBP against Lung Cancer Cells, network pharmacology and experimental verification were combined. Molecular docking elucidated the binding conformation and mechanism of the primary active components (Osmundacetone and hispidin) to the core targets CASP3, PARP1 and TP53. In addition, potential molecular mechanisms of PBP predicted by network pharmacology analysis were validated in vitro. PBP significantly inhibited the human lung cancer A549 cells and showed typical apoptotic characteristics, without significant cytotoxicity to normal human embryonic kidney (HEK293) cells. Analysis using flow cytometry and western blot indicated that PBP caused apoptosis, cell cycle arrest, reactive oxygen species (ROS) accumulation, and mitochondrial membrane potential (MMP) depression in A549 cells to exercise its antitumor effects. These results reveal that PBP has great potential for use as an active ingredient for antitumor therapy.

## 1. Introduction

Cancer significantly affects public health globally [1]. However, the efficiency of many traditional tumor treatments, such as surgery, radiotherapy and chemotherapy, is not ideal due to their severe side effects. Thus, new therapeutic approaches are urgently needed. In recent years, natural products, as a valuable source of antitumor compounds, have gained increasing attention due to their few adverse effects. Many studies have suggested that polyphenols, which are generally recognized as antioxidants, possess antitumor and proapoptotic effects [2,3]. Polyphenols from the pinecones of Pinus koraiensis significantly inhibited S180 cell growth by activating the mitochondrial apoptotic pathway [4]. It has been demonstrated that polyphenols from green tea have prophylactic and therapeutic effects against tumors [5]. Resveratrol, a nonflavonoid polyphenol present in grapes, soybeans, peanuts, etc., displays therapeutic potential in breast cancer treatment [6]. Resveratrol can also inhibit murine hepatocellular carcinoma by downregulating CD 8+, CD 122+, Tregs cells [7]. However, the antitumor efficiency of polyphenols remains to be explored and utilized.

*Phellinus baumii* (*P. baumii*) is a well-known medicinal fungus belonging to *Phellinus*, *Hymenochaetaceae*, *Aphyllophorales*, *Hymenomycetes*, and *Basidiomycotas*. In China, the fruiting bodies of *P. baumii* have been used as a traditional medicine for centuries [8]. *Phellinus* spp. show a wide range of pharmacological activities, especially anticancer properties, and have the reputation of ‘forest gold’ due to their abundant components, including polysaccharides, polyphenols, terpenoids, steroids, etc. The polysaccharide extract of *P. baumii* showed toxicity to tumor cells in vitro [9,10] and was found to alleviate ulcerative colitis [11] and high-fat high-fructose diet-induced insulin resistance in mice [12]. The terpenoids produced by *Phellinus* spp. were found to enhance immune function [13,14]. The ethanol extract of *P. baumii* exerted cytotoxic activity against many kinds of human tumor cell lines [15]. Polyphenols of *Phellinus* spp. have been found to have anti-inflammatory and antidiabetic effects on diabetic mice [16,17]. However, whether *P. baumii* polyphenols have antitumor effects is unclear.

In recent years, network pharmacology has become one of the frontiers and hotspots in the field of traditional Chinese medicine research [18]. Network pharmacology research is based on high-throughput omics data analysis, computer virtual computing and network database retrieval. It is a good bioinformatics method to reveal the therapeutic effect of natural drugs with complex components on diseases.

In this study, we identified the components of polyphenol extract from *Phellinus baumii* (PBP) by UPLC–ESI–QTOF–MS for the first time. Additionally, we integrated network pharmacology with experimental verification to clarify the possible mechanism of against lung cancer cells effect of PBP.

## 2. Results

### 2.1. Characterization of the Chemical Constituents of PBP

The identification of the components of PBP was performed using UPLC–ESI–QTOF–MS. The base peak chromatograms of PBP in negative ion mode are shown in Figure 1a. A total of 17 peaks were tentatively identified (Table 1). Four peaks (peak 1, 2, 5 and 7) were confirmed qualitatively with a mixed standard of protocatechuic aldehyde (PubChem CID: 8768), caffeic acid (PubChem CID: 689043), osmundacetone (PubChem CID: 9942292) and hispidin (PubChem CID: 54685921) (Figure 1b). Eight compounds were previously reported to exist in the *P. baumii*, including phellibaumin B (peak 4, PubChem CID: 53248680) [19], hypholomine B (peak 6, PubChem CID: 54730031) [20,21], interfungin B (peak 8, PubChem CID: 54718836) [19,22], phelligridimer A (peak 10, PubChem CID: 16155688) [19,23], davallialactone (peak 12, PubChem CID: 54715402) [20,24], phellibaumin E (peak 14, PubChem CID: 54738502) [19,25], phelligridin I (peak 16, PubChem CID: 54689623) [26,27], and inoscavin A (peak 17, PubChem CID: 10434469) [20,24]. Then, five compounds were tentatively identified according to the UPLC—MS results, including kielcorin (peak 3, PubChem CID: 13834128) [28], sterubin (peak 9, PubChem CID: 4872981) [29], (E)-4-(4-hydroxyphenyl)-3-buten-2-one (peak, PubChem CID: 11796857) [30], citrinin (peak 13, PubChem CID: 54680783) [31] and 3-(4,6-Dihydroxy-2-oxochromen-3-yl)-8-hydroxy-2-methoxy-2,3-dihydrofuro [3,2-c] chromen-4-one (peak 15, PubChem CID: 87782626) [32] (Appendix A). There were two unknown peaks on the chromatogram with retention times of 9.53 and 11.53 min. The formula of peak N1 is C10H10O5, and peak N2 was determined to be an isomer of hypholomine B. The 2D chemical structures of PBP compounds are shown in Table 1.

### 2.2. Identification of Potential Targets of PBP in Lung Cancer Treatment

In total, 221 potential targets of PBP components were collected from the Drugbank and Traditional chinese medicine systems pharmacology database and analysis platform (TCMSP) databases with 17 components of PBP as keywords (Appendix A). A total of 1035 potential lung cancer targets were identified from the OMIM, GeneCards, DisGeNET, and Treatment targets database (TTD) databases (Figure 2a). The intersection of PBP component targets and lung cancer related targets resulted in 60 common targets, which were used as potential targets for subsequent research (Figure 2b and Appendix A).

### 2.3. Components-Disease-Targets Interaction Network

To clearly understand the key components of PBP in the treatment of lung cancer, the active Components-Disease-Targets network was visualized using Cytoscape 3.9.1 (Figure 2c). The network contains 74 nodes and 193 edges. The degree value was obtained by CytoNCA tool and ranked in descending order. According to network analysis, compounds have an average degree of 11.0, excluding 5 components with a degree value of 0. The active components greater than the average are shown in Table 2: protocatechuic aldehyde (degree = 26), caffeic acid (degree = 25), osmundacetone (degree = 22), hispidin (degree = 20), citrinin (degree = 15), davallialactone (degree = 11). Therefore, the 6 compounds are considered to be potential bioactive compounds of PBP against lung cancer.

### 2.4. Gene Ontology (GO) and Kyoto Encyclopedia of Genes and Genomes (KEGG) Enrichment Analysis

A total of 312 GO terms with *p* < 0.05, including 214 BP, 43 CC, and 55 MF, were considerably enriched (Appendix A). Visual representations of the top 10 considerably enriched terms in the BP, CC, and MF categories were shown (Figure 3a). Furthermore, a total of 150 pathways with *p* < 0.05 were identified (Appendix A). The top 20 pathways were displayed in (Figure 3b). The 33 intersection targets were related to ‘pathway in cancer’. Combined with ‘pathway in cancer’ (Appendix A), it can be shown that these potential targets in the ‘pathway in cancer’ are mainly focused on apoptosis and cell cycle-related pathways. The results show that PBP may treat lung cancer by regulating apoptosis and cell cycle progression.

### 2.5. Protein-Protein Interaction (PPI) Network and Core Targets

A PPI network comprising 60 intersectional targets was created using the STRING database (interaction score > 0.9) to further examine the potential mechanism by which PBP treats lung cancer (Figure 4a). Then the protein interaction results of STRING analysis were visualized by Cytoscape 3.9.1 (Figure 4b). This PPI network is ranked by degree score. Stronger protein interaction was indicated by a higher score. The 6 core targets were obtained: TP53, STAT3, PARP1, HMOX1, AKT1, CASP3. Among them, TP53, PARP1, CASP3 targets are key genes in the pathway in cancer, and are associated with apoptosis and cell cycle progression. PBP may treat lung cancer by acting on these core targets.

### 2.6. Molecular Docking

Autodock Vina was used to verify the bindings of TP53, PARP1, and CASP3 to the six key active compounds of PBP. The binding energies indicates the degree of complementarity between the components and the targets. The lower the binding energy, the higher the stability. The molecular dockings of Rotocatechuic aldehyde, caffeic acid with TP53, PARP1, CASP3 were failed. The binding energies of osmundacetone with CASP3, PARP and TP53 quercetin and wogonin were −6.4, −7.2, −5.3 kcal/mol, respectively. The binding energies of hispidin with CASP3, PARP and TP53 were −6.8, −7.1, −6.2 kcal/mol, respectively (Table 2). The results of degree and binding energy showed that osmundacetone and hispidin were the main active compounds of PBP treats lung cancer. Next, the molecular docking results of autodock vina were treated with PyMol. As shown in Figure 4c, osmundacetone forms 4 hydrogen bonds with TYR-276, MET-39 and TYR-37 in CASP3, hispidin formed 2 hydrogen bonds with MET-39 and TYR-37 in CASP3. In addition, osmundacetone forms 5 hydrogen bonds with GLY-888, GLU-988, GLY-863, SER-904 and TRP-861 in PARP, hispidin formed 3 hydrogen bonds with GLN-853 and GLU-840 in PARP, osmundacetone forms 1 hydrogen bonds with LEU-330 in TP53, hispidin formed 7 hydrogen bonds with ARG-335, GLY-334, ILE-332, leu-330, THR-329 in TP53 (Figure 4c).

### 2.7. PBP Significantly Inhibited the Viability of A549 Cells

The toxicity of PBP to human tumor cells and normal cells was examined using CCK-8 assays. According to the results, the IC_50_ values against A549 were 49.07 ± 0.5 μg/mL and varied by dose and duration (Figure 5a). In addition, we examined the toxicity of PBP to various human tumor cells. After 48 h treatment of PBP, the IC_50_ values against Hela, HepG2, T24 and HCT116 cells were 96.14 ± 4.9, 140.1 ± 9.6, 105.0 ± 10.7 and 54.05 ± 0.7 μg/mL, respectively (Table 3, Figure A1). However, at doses below 80 μg/mL, PBP had no discernible toxic effects on HEK293 cells (normal cells, not tumor cells) (Figure 5b). These results indicate that PBP is significantly toxic to human lung cancer A549 cells, but not to normal human HEK293 cells.

Cell morphology was examined using phase-contrast microscopy to further evaluate the effects of PBP on A549 cells. The A549 cells in the control group were polygonal or fusiform and adhered to the plate well. The A549 cells treated with PBP, however, displayed morphological features such as shrinkage and rounding (Figure 5b). The number of A549 cells dramatically decreased as PBP concentration rose. Under treatment with 80 μg/mL PBP, the shape of the A549 cells became irregular, and floating cells were clearly observed, forming a smooth contour and separating from the surroundings. Consequently, PBP may induce cell death through certain pathways and impair cell proliferation, which needs to be further confirmed.

### 2.8. PBP Induced Apoptosis in A549 Cells

Flow cytometry was used to identify cell apoptosis using Annexin V-FITC and PI staining methods. Annexin V−/PI+ (upper left quadrant) represents cell debris or other causes of cell death cells. Annexin V−/PI− (lower left quadrant) represents normal living cells. Annexin V+/PI+ (upper right quadrant) represents late apoptotic cells. Annexin V+/PI− (lower right quadrant) represents early apoptotic cells. The results showed a rise in the total apoptotic rate of A549 cells from 5.20% ± 1.5% to 9.72% ± 0.28%, 18.46% ± 1.93% and 25.42% ± 0.86%, respectively. Notably, the early apoptotic rate increased from 1.3% ± 0.56% to 7.37% ± 0.87%, 18.80% ± 2.33% and 32.88% ± 1.64%, and the late apoptotic rate increased from 3.47% ± 0.71% to 6.56% ± 0.67%, 6.72% ± 0.83% and 8.26% ± 1.54%, respectively (Figure 6a,c). Annexin V+ in Figure 6b also confirmed the result that PBP induced A549 cell apoptosis. The results suggested that PBP may influence A549 cell viability by inducing apoptosis in a dosage manner.

The B-cell lymphoma-2 (Bcl-2) and Bax protein family regulates apoptosis and influences mitochondrial function [33]. Various downstream apoptotic mediators are activated and participate in a series of cell death cascade reactions. According to western blot analysis, PBP dose-dependently increased the proapoptotic protein Bax while downregulating the antiapoptotic regulator Bcl-2 (Figure 6e). Changes in Bcl-2 and Bax usually indicate that apoptosis occurs through the mitochondria-dependent apoptotic pathway. As a result, the expression levels of additional regulators involved in the mitochondrial apoptotic pathway were examined. The expression of P53, apoptosis inducing factor (AIF), apoptotic protease activating factor-1 (Apaf-1), Cytochrome C, active caspase-3, active caspase-9, and active caspase-8 were all enhanced by PBP, but inactive zymogen proteins including pro-caspase-9, pro-caspase-8, and pro-caspase-3 were decreased. Downregulated PARP expression and upregulated cleaved PARP expression also suggested that PBP promotes PARP cleavage in A549 cells (Figure 6e). These results suggest that PBP may release apoptosis activators by altering mitochondrial function, thereby activating the caspase pathway cascade. The upstream initiators pro-caspase-9 and pro-caspase-8 become activated to form active caspase-9 and active caspase-8, which in turn activate pro-caspase-3 to become active caspase-3, which acts as an executor to induce PARP cleavage, ultimately triggering endogenous apoptosis and DNA fragmentation. Protein expression histograms displaying the amounts of apoptosis-related proteins in Appendix A. As a result, PBP could cause apoptosis in A549 cells by activating the intrinsic apoptotic pathway of the mitochondria.

### 2.9. PBP Induced Mitochondrial Membrane Potential (MMP) Depolarization in A549 Cells

Apoptosis is associated with mitochondrial dysfunction and MMP depolarization [34]. We examined the changes in MMP using Mitochondrial Membrane Potential and Apoptosis Detection Kit. Annexin V−/CMXRos+ represents normal living cells. Annexin V+/CMXRos- indicates cells with apoptosis and decreased mitochondrial membrane potential. The considerable decrease in fluorescence intensity of the Mito-Tracker Red CMXRos probe suggested that the number of A549 cells with normal MMP was dramatically decreased after treatment with PBP. The number of apoptotic cells labeled with the green fluorescent probe Annexin V-FITC increased significantly, again verifying that PBP treatment induced A549 cell apoptosis (Figure 6b,d). The blue fluorescent probe Hoechst 33342 was used for fluorescence detection. Compared to the control group, apoptotic cells with Annexin V+/CMXRos- had smaller nuclei, and part of these nuclei had decomposed into fragments. According to these results, PBP-induced apoptosis in A549 cells may be related to MMP depolarization.

### 2.10. PBP Arrested the Cell Cycle of A549 Cells in S Phase

Following 48 h of PBP administration, 20,000 cells from each sample were gathered for flow cytometric analysis. The percent of A549 cells in S phase rose significantly after treatment with 80 g/mL PBP, rising from 9.35% 1.10% to 24.67% 0.33%, whereas the percentage in G1 stage decreased from 82.94% ± 1.43% to 67.90% ± 0.40%. However, PBP administration had no discernible impact on the percentage of A549 cells that were in the G2 phase (Figure 7a,c). These results suggested that PBP was able of S phase arrest in A549 cells.

Western blotting was used to confirm the expression levels of important cell cycle-related proteins. The results indicated that Rb was phosphorylated (P-Rb) under PBP treatment, resulting in increased release of free E2F transcription factor 1 (E2F1) from the E2F1/Rb complex. Similarly, PBP treatment inhibited Cyclin Dependent Kinase 2 (CDK2) and cyclin A2 and promoted P21 expression (Figure 7c). Protein expression histograms displaying the amounts of apoptosis-related proteins in Appendix A. The expression levels of the target proteins were normalized to those of β-actin. Therefore, PBP arrested A549 cells in S phase by altering the expression of key regulators, such as cyclin A2/CDK2, P21 and P53.

### 2.11. PBP Promoted Reactive Oxygen Species (ROS) Generation in A549 Cells

ROS are implicated in the apoptosis process [35,36]. The 10,000 cells were collected for each DCFH-DA staining sample after treatment with 0, 20, 40 and 80 g/mL PBP. Intracellular ROS can oxidize nonfluorescent DCFH to fluorescent DCF, which can be detected by flow cytometry [37]. The histograms of A549 cells shifted to the right when the PBP concentration increased. After treatment with 80 μg/mL PBP, the geomean fluorescence intensity of DCF increased remarkably from 52.90 ± 1.56 to 85.13 ± 2.30 in A549 cells (Figure 7a,c). The results These results indicated that PBP-induced apoptosis may be associated with ROS production. that PBP-induced apoptosis ould be associated with ROS production.

### 2.12. PBP Induced Mitochondrial Membrane Potential (MMP) Depolarization in A549 Cells

Apoptosis is associated with mitochondrial dysfunction and MMP depolarization [34]. We examined the changes in MMP using an MMP kit. The considerable decrease in fluorescence intensity of the MitoTracker Red CMXRos probe suggested that the number of A549 cells with normal MMP was dramatically decreased after treatment with PBP. The number of apoptotic cells labeled with the green fluorescent probe Annexin V-FITC increased significantly, again verifying that PBP treatment induced A549 cell apoptosis (Figure 7b,d). The blue fluorescent probe Hoechst 33342 was used for fluorescence detection. Compared to the control group, apoptotic cells without red fluorescence (MMP depolarization) but with green fluorescence (phosphatidylserine outside the cell membrane) had smaller nuclei, and part of these nuclei had decomposed into fragments. According to these results, PBP-induced apoptosis in A549 cells may be related to MMP depolarization.

## 3. Discussion

*P. baumii*, a traditional herbal medicine, has shown antitumor properties against a wide range of cancers [38]. Natural products are essential sources for the research and development of novel anticancer medicines. In this work, we determined the composition of PBP and identified its 17 chemical components and 2D structures, many of which have been proved to exert various pharmacological activities. For instance, hispidin induced apoptosis mediated by ROS in colon cancer cells, and could enhance the therapeutic activities of gemcitabine against pancreatic cancer stem cells [39,40].

Through the network pharmacology method, a total of 60 targets obtained by intersection were regarded as potential targets for PBP treatment of lung cancer. Protocatechuic aldehyde, caffeic acid, osmundacetone, hispidin, citrinin, davallialactone are considered to be potential bioactive compounds of PBP against lung cancer. GO and KEGG analysis showed the 33 of 61 (54%) intersection targets are related to apoptosis and cell cycle pathways (Figure 3). The results show that PBP may treat lung cancer by regulating apoptosis and cell cycle progression. Molecular docking elucidated the binding conformation and mechanism of the primary active components (Osmundacetone and hispidin) to the core targets CASP3, PARP1 and TP53 (Figure 4). 

Apoptosis, a genetically regulated process of programmed cell death, has been acknowledged as a new cancer therapy [9,41]. In this study, typical apoptotic characteristics were seen in PBP-treated A549 cells, including cell volume shrinkage, detachment from the surrounding cells (Figure 5c), MMP depolarization (Figure 5b), positive Annexin V-FITC staining (Figure 5a,b). Other studies have shown that some bioactive substances in natural products exert antitumor effects mainly through three pathways, including the intrinsic mitochondrial pathway, extrinsic death receptor pathway and endoplasmic reticulum stress pathway [42]. Figure 5 indicated that PBP may induce apoptosis through mitochondrial department apoptosis pathways. The changes in Bcl-2 and Bax resulted in the disappearance of the MMP, therefore increasing the release of cytochrome C and AIF (Figure 5e), which activate a cascade of caspases. Active caspase-9 and active caspase-8 further activates downstream caspase-3, which acted as an executor to cleave PARP, resulting in irreparable DNA damage, thereby inducing apoptosis in A549 cells.

Therapeutics that target the cell cycle have emerged as a successful method for the treatment of cancer [43]. Cell cycle regulation is dependent on the cyclin family and CDK family of proteins. A reduction in or dysregulation of the cyclin A/CDK2 complex could cause cells to arrest in S phase. In this study, A549 cells treated with PBP were arrested in S phase (Figure 6a,e). Furthermore, PBP resulted in Rb phosphorylation (P-Rb), increasing the release of E2F1 from the E2F1/Rb complex. Moreover, PBP treatment inhibited CDK2 and cyclin A2 and promoted the expression of P21. These results suggest that inhibition of cyclin A2/CDK2 and upregulation of P21 and P53 play key roles in PBP-induced S-phase arrest in A549 cells (Figure 5e and Figure 6e).

Previous studies have shown that ROS can be associated with a variety of cell cycle, apoptosis, autophagy pathway [44,45]. In this study, PBP induced ROS production and MMP depolarization (Figure 5b), indicating that it is a potential factor to promote apoptosis and S phase arrest in A549 cells.

In conclusion, our investigations demonstrated that PBP exhibited remarkable antitumor capacities against lung cancer A549 cells in vitro. PBP induces apoptosis, cell cycle arrest, ROS accumulation and MMP depolarization in A549 cells (Figure 8). Moreover, the against lung cancer effects of PBP in vivo were not investigated in this work. However, it has been demonstrated that hot water extract of *Phellinus linteus* has been demonstrated to significantly suppress the S180 melanoma sarcoma and mouse colon cancer in vivo [46]. In addition, Jae-Sung Bae et al. found that polysaccharides isolated from *Phellinus gilvus* inhibit melanoma growth in mice [47]. What these in vivo studies have in common is that they all use intragastric administration. It is reasonable to speculate that the anti-tumor effect of *Phellinus linteus* is achieved by improving the changes of intestinal microflora, which deserves further consideration and research. These results in vivo suggest that PBP may have great potential for anti-tumor effects in vivo, including lung cancer. The polyphenol extract of *P. baumii* could be a potential candidate for future antitumor drug development, and it also lays a foundation for the study of the anti-tumor mechanism of *P. baumii* polyphenols.

## 4. Materials and Methods

### 4.1. Reagents

Dulbecco’s modified essential medium (DMEM, high glucose) and fetal bovine serum (FBS) were purchased from Gibco Industries Inc. (Grand Island, NY, USA). Dimethyl sulfoxide (DMSO) was purchased from Sigma—Aldrich (St. Louis, MO, USA). An Annexin V FITC apoptosis detection kit was obtained from Dojindo Molecular Technologies (Kumamoto, Japan). Trypsin (0.25% with or without EDTA), CCK-8, cell cycle detection kit and reactive oxygen species (ROS), Mitochondrial Membrane Potential Detection Kit (C1071; Beyotime, Shanghai, China) assay kit were purchased from Beyotime Institute of Biotechnology (Jiangsu, China). The fruiting bodies of *P. baumii* were obtained from Huqingyutang drugstore (Hangzhou, China).

HPLC-grade methanol was purchased from Sinopharm Chemical Reagent Co., Ltd. (Shanghai, China). Protocatechuic aldehyde, caffeic acid, osmundacetone and hispidin (all HPLC grade) were obtained from Weikeqi Biological Technology Co., Ltd. (Chengdu, China).

### 4.2. Preparation of PBP

The fruiting bodies of *P. baumii* were dried, ground, and passed through an 80-mesh sieve. The powder was suspended in 60% ethanol (1:60, *w*/*v*). After 30 min of ultrasound treatment, reflux extraction was performed at 90 °C for 1 h. The extract was subjected to vacuum filtration, and the filtrate was collected and centrifuged (10,000 rpm, 5 min). The solvent of the supernatant was evaporated at 40 °C in a rotary evaporator under low pressure and then the remaining material was lyophilized with a freeze dryer (LABCONCO, MO, USA). The crude ethanolic extract was purified with an AB-8 macro porous adsorption resin column (Solarbio, Beijing, China) eluted using distilled water, 30% ethanol and 70% ethanol in sequence. Then, the 70% ethanol eluent was collected, concentrated and lyophilized to obtain the purified PBP. The total polyphenol content of PBP was 94.47% ± 4.10%, as determined by the Folin-Ciocalteu (Solarbio, Beijing, China) method [48]. The obtained PBP powder was stored at 4 °C for the following experiments.

### 4.3. Identification of PBP Compounds by UPLC–ESI–QTOF–MS

PBP was dissolved in 80% methanol at a final concentration of 2 mg/mL. A Waters UPLC (Waters Corp., Milford, MA, USA) with a Waters ACQUITY UPLC HSS T3 column (150 mm × 3.0 mm i.d., 1.7 µm) was used in the assay at a temperature of 50 °C and flow rate of 0.3 mL/min. The injection volume of the sample was 3 μL, and the UV detector was set at 254 nm. The mobile phases were 0.1% formic acid-water (A) and 0.1% formic acid-acetonitrile (B). The linear gradient was as follows: 0/10, 2/20, 25/35, 35/95 (min/B%). Mass spectrometry was performed using an AB Triple TOF 5600 plus System (AB SCIEX, Framingham, USA). The optimal MS conditions were as follows: the scan ranges (m/z) of precursor ions and product ions were set as 100–1500 Da and 50–1500 Da, respectively; the source voltage and temperature for negative ion mode were −4.5 kV and 550 °C, respectively, and those for positive ion mode were +5.5 kV and 600 °C, respectively; the pressures of gas 1 (air) and gas 2 (air) were set to 50 psi, and that of the curtain gas (N2) was set to 35 psi; the maximum permissible error was set to ± 5 ppm; and the declustering potential (DP) and collision energy (CE) were 100 V and 10 V, respectively. MS/MS acquisition was performed with almost the same parameters except that the CEs were set to −40 ± 20 V and 40 ± 20 V in negative and positive ion modes, respectively. The mass axis was calibrated with a CDS pump to reduce the mass axis error to less than 2 ppm.

The data were analyzed with PeakView 1.2 software (ABSciex, Vaughan, ON, Canada). A compound library of polyphenol constituents previously discovered in *Phellinus* spp. was constructed to help with identification. Published research and chemical databases, such as Reaxys, PubChem, and ChemSpider, also helped with the identification of these compounds.

### 4.4. Prediction of Potential Targets of PBP during Lung Cancer Treatment

Potential targets information of 17 active compounds of PBP were collected from TCMSP database (https://old.tcmsp-e.com/tcmsp.php (accessed on 10 November 2022)) [49] and Drugbank database (https://go.drugbank.com/ (accessed on 10 November 2022)) [50]. Lung cancer-treat targets were collected from 4 databases using the keyword ‘lung cancer’: GeneCards database (https://www.genecards.org/ ((accessed on 10 November 2022))) [51], Human Online Mendelian Inheritance database (OMIM, https://www.omim.org/ ((accessed on 10 November 2022))) [52], DisGeNET database (https://www.disgenet.org/ ((accessed on 10 November 2022))) [53] and TTD (http://db.idrblab.net/ttd ((accessed on 10 November 2022))) [54]. The intersection of PBP active component targets and lung cancer-related targets is regarded as the potential targets for PBP treatment of lung cancer. Visualization using Venny 2.1 (https://bioinfogp.cnb.csic.es/tools/venny ((accessed on 10 November 2022))).

### 4.5. Component-Disease-Target Interaction Network

The potential targets of active compounds in PBP were constructed as a Drug-Component-Disease-Target visual interaction network using Cytoscape 3.9.1 software [55]. Compounds with more than the mean of degree centrality are considered core compounds.

### 4.6. GO and KEGG Enrichment Analysis

DAVID database (https://david.ncifcrf.gov/ ((accessed on 10 November 2022))) [56] was used to further analyze the potential targets of PBP in the treatment of lung cancer. We used DAVID to analyze the biological process (BP), cellular component (CC), and molecular function (MF) terms of the common targets. The main pathways that were enriched were subsequently examined using the KEGG database (https://www.kegg.jp/ ((accessed on 10 November 2022))) [57]. The KEGG pathways and GO terms were determined with *p* < 0.05.

### 4.7. PPI Network and Core Targets

The PBP treatment of lung cancer intersection targets were analyzed by STRING database (https://string-db.org/ ((accessed on 10 November 2022))) [58]. The species was set to ‘homo’, the confidence score was set to >0.9 to obtain a more relevant protein interaction network. The visualized PPI network and topological values was constructed by Cytoscape 3.9.1. Targets that simultaneously satisfy the mean values of degree centrality (DC), betweenness centrality (BC), and closeness centrality (CC) were identified as core targets.

### 4.8. Molecular Docking

The 3D chemical structures of the PBP components were obtained from the PubChem database (https://pubchem.ncbi.nlm.nih.gov/ ((accessed on 10 November 2022))). The 3D structures of the core targets from the protein database (PDB, https://www.rcsb.org/ ((accessed on 10 November 2022))) [59]. Autodock 4, Autodock Tools 1.5.7 and Autodock Vina [60] are used to detect and select rotatable bonds in compounds, remove water, add hydrogen from protein 3D structures. Then, Autodock Vina was used for molecular docking of PBP compounds with the core targets. The binding activity between each component and the target was evaluated according to the binding energy, the affinity <−5 kcal/mol was defined as a good binding activity. PyMOL software was used for analysis and plotting.

### 4.9. Cell Lines and Culture

Human lung cancer A549 cell line (RRID: CVCL_0023) and human embryonic kidney HEK293 cell line (RRID: CVCL_0045) were obtained from the Cell Research Institute of the Chinese Academy of Sciences (Shanghai, China). The cells used in the experiment were 6–10 generation cell lines. The cells were cultured in DMEM (high glucose) supplemented with 10% (*v*/*v*) FBS at 37 °C under a humid atmosphere with 5% CO_2_.

### 4.10. Cell Viability Assay and Morphological Observations

Cells (5 × 10^3^ cells/well) were seeded in 96-well plates (Costar Corning, Rochester, NY, USA), cultured for 24 h to allow for adherence, and then treated with PBP for 24, 48 and 72 h. After that, 10 μL of CCK-8 solution was added to each well, and the incubation continued for 2 h at 37 °C. The absorbance was measured using a multimode reader (Thermo Electron Corporation, MA, USA) at 450 nm. The wells without cells but with all other reagents were used as the blank, and the cell viability rate was calculated as follows:Cell Viability (%) = (A450 _treated_ − A450 _blank_)/(A450 _Control_ − A450 _blank_) × 100%

The cell morphology was observed and photographed by an Olympus phase contrast microscope (Olympus, Tokyo, Japan).

### 4.11. Cell Apoptosis Assay

Cell apoptosis was determined using an Annexin V, FITC Apoptosis Detection Kit and flow cytometer. A549 cells were seeded in 6-well plates at 1 × 10^5^ cells/well. After 24 h of culture, the cells were treated with PBP (0, 20, 40, 80 μg/mL) for 48 h. Then, the cells were digested, washed and stained with Annexin V-FITC (5 μL/well) and PI (5 μL/well), incubated for 15 min in the dark, and analyzed on a BD FACSVerse flow cytometer. The cell apoptosis rate data were analyzed using FlowJo software.

### 4.12. MMP Assay

The membrane potential was determined by using a Mitochondrial Membrane Potential and Apoptosis Detection Kit with Mito-Tracker Red CMXRos and Annexin V-FITC (C1071; Beyotime, Shanghai, China). Cells were seeded into 6-well plates (1 × 10^5^ cells/well) and cultured for 24 h. After 48 h of incubation with PBP, the cell culture medium was removed by aspiration and the cells were washed once with PBS. Then, Annexin V-FITC binding solution (188 μL/plate), Annexin V-FITC (5 μL/plate), MitoTracker Red CMXRos staining solution (2 μL/plate), and Hoechst 33342 staining solution (5 μL/plate) were added. The samples were incubated in the dark for 20–30 min at room temperature and then placed in an ice bath. The sample smears were visualized under a laser confocal microscope (Leica Microsystems, Wetzlar, Germany).

### 4.13. Western Blot Assay

Cells were seeded into 6-well plates (1 × 10^5^ cells/well) and cultured for 24 h. After treatment with PBP (20, 40, 80 μg/mL) for 48 h, the total proteins were extracted by Cell lysis buffer for Western and IP (Beyotime, Shanghai, China). The concentrations of extracted protein were determined by using a BCA protein assay kit (Beyotime, Shanghai, China). Proteins were loaded into each well of a 10% or 15% sodium dodecyl sulfate—polyacrylamide gel electrophoresis (SDS—PAGE) and then transferred to a polyvinylidene difluoride (PVDF) membrane. After blocking with 5% nonfat dry milk with 0.05% Tween 20 in TBS (1×) for 1 h at 25 °C, the membranes were incubated with the corresponding primary antibodies at 4 °C overnight. The primary antibodies were rabbit anti-human P53 (E26, Abcam, Cambridge, UK), β-actin (R1207-1, HUABIO, Woburn, MA, USA), AIF (ET1603-4, HUABIO), Bcl-2 (ER0602, HUABIO), Bax (ET1603-34, HUABIO), Apaf-1 (R1312-20, HUABIO), poly ADP-ribose polymerase (PARP, ET1608-56, HUABIO), cleaved PARP (ET1608-10, HUABIO), CDK2 (ET1602-6, HUABIO), phosphorylated retinoblastoma protein (P-Rb, ET1602-36, HUABIO), cyclin A2(ET1612-26, HUABIO), E2F transcription factor 1 (E2F1, ET1701-73, HUABIO), active + pro caspase 9 (R1308-12, HUABIO), caspase-3 (ET1602-39, HUABIO), active caspase-3 (ET1602-47, HUABIO), caspase-8 (ET1603-16, HUABIO), cytochrome C(ET1610-60, HUABIO). The membranes were then incubated with secondary antibodies, HRP Conjugated Goat anti-Rabbit IgG (HUABIO, Hangzhou, China), at room temperature for 1 h. Protein signals were visualized by Enhanced ECL Chemiluminescent Substrate Kit (Yeasen, Shanghai, China).

### 4.14. Cell Cycle Assay

Cell cycle analysis was conducted by flow cytometry. Cells were seeded into 6-well plates (2 × 10^5^ cells/well) and cultured for 24 h. After treatment with PBP for 48 h, the cells were digested with trypsin, rinsed twice with cold phosphate-buffered saline (PBS), fixed with 70% precooled ethanol and washed three times with PBS. Then, the cells were incubated with PI staining solution containing RNase A at 37 °C for 1.5 h. Cell cycle analysis was conducted on a BD FACSVerse flow cytometer and analyzed with Modfit LT software.

### 4.15. ROS Generation Assay

Cells were seeded into 6-well plates (1 × 10^5^ cells/well) and cultured for 24 h. After 48 h of incubation with PBP, the cells were collected and washed twice with PBS. Then, the cells were incubated with DCFH-DA at 37 °C for 20 min, rinsed twice and resuspended in DMEM. The ROS levels in the cells were determined with a BD FACSVerse flow cytometer and the data were analyzed by FlowJo software.

### 4.16. Statistical Analysis

Statistical analyses were performed using GraphPad Prism software version 8.0.2 (GraphPad, San Diego, CA, USA). The statistical significance of the differences between groups was analyzed by one-way ANOVA with Dunnett’s test as a post hoc analysis. Data are presented as the mean ±SD. Differences with a *p* value < 0.05 or *p* value < 0.01 were considered statistically significant or extremely significant, respectively. The flow cytometry gating strategy is shown in (Figure A2).

## Figures and Tables

**Figure 1 ijms-23-16141-f001:**
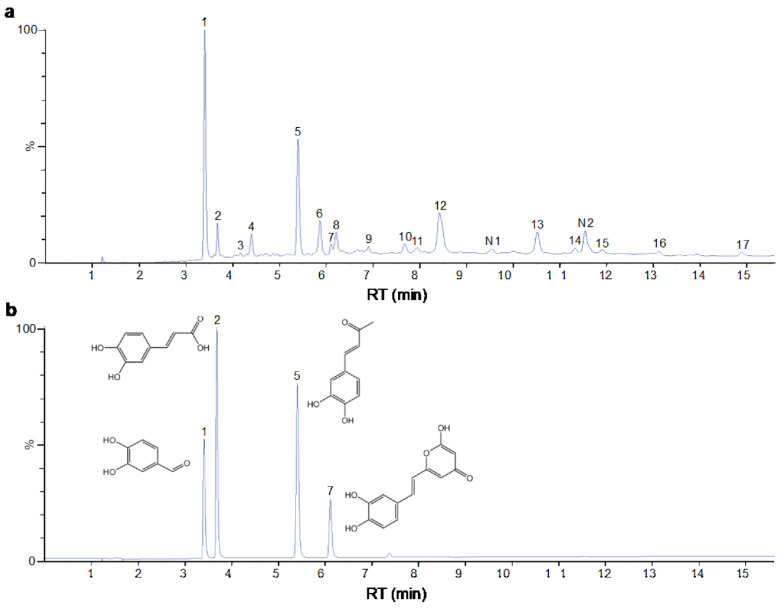
(**a**) UV chromatogram of PBP at 254 nm. (**b**) UV chromatogram of mixed standards of protocatechuic aldehyde, caffeic acid, osmundacetone and hispidin.

**Figure 2 ijms-23-16141-f002:**
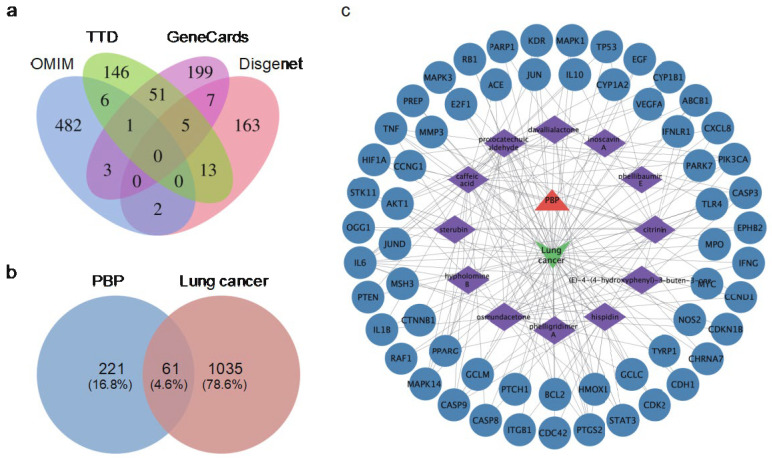
The targets of PBP treat lung cancer and the Components-Disease-Targets network. (**a**) Venn diagram of lung cancer targets; (**b**) Venn diagram of PBP treat lung cancer; (**c**) Drug-Components-Disease-Targets network. The orange triangle represents PBP. The green wedge represents lung cancer. The active substances in PBP are represented by the purple prisms. The targets are shown as blue circles.

**Figure 3 ijms-23-16141-f003:**
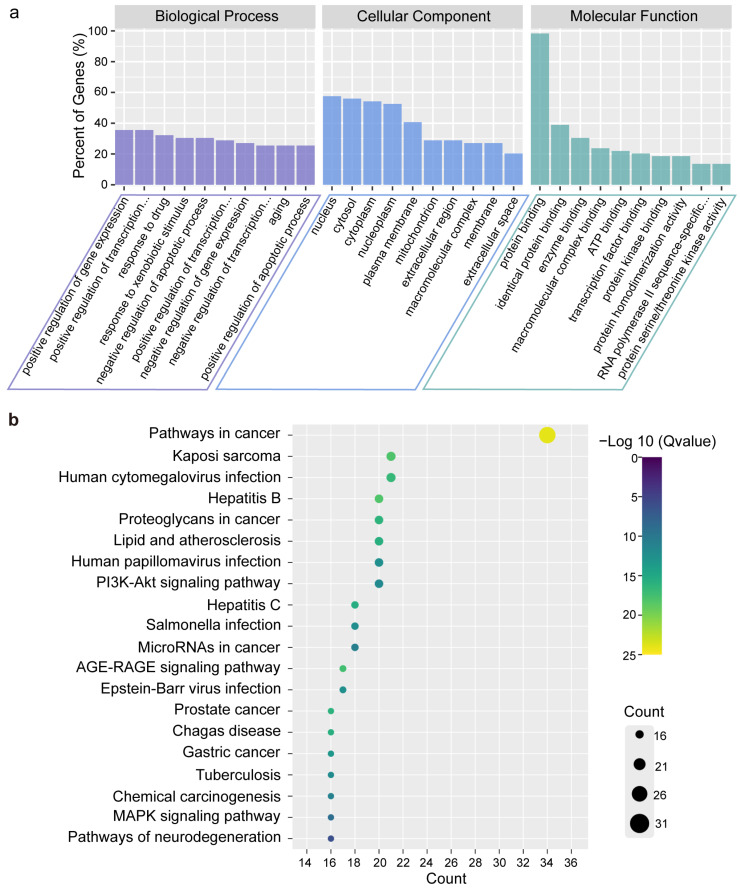
Analysis of GO and KEGG enrichment to identify the anti-tumor mechanisms. (**a**) The top 10 in BP terms, CC terms, and MF terms of the GO terms (*p* < 0.05); (**b**) The top 20 KEGG pathways (*p* < 0.05).

**Figure 4 ijms-23-16141-f004:**
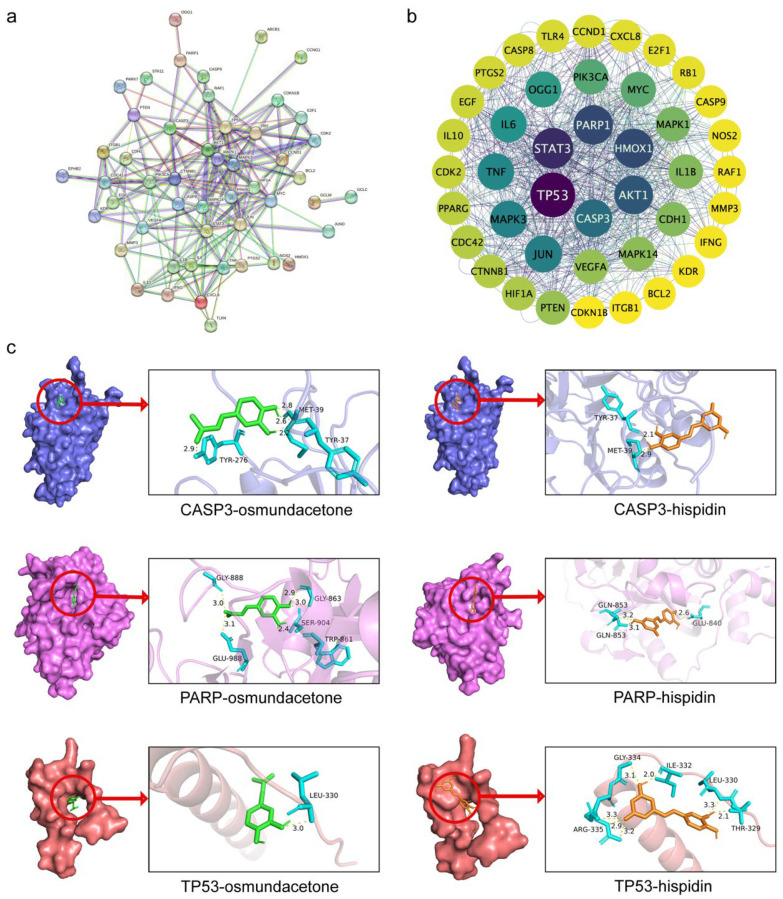
Identification and molecular docking verification of PBP anti-lung cancer core targets. (**a**) PPI network diagram obtained by STRING database analysis (interaction score > 0.9); (**b**) Visualizing the PPI network was completed using Cytoscape 3.9.1; (**c**) The docking mode of CASP3- osmundacetone, PARP-osmundacetone, TP53-osmundacetone, CASP3-hispidin, PARP-hispidin and TP53-hispidin.

**Figure 5 ijms-23-16141-f005:**
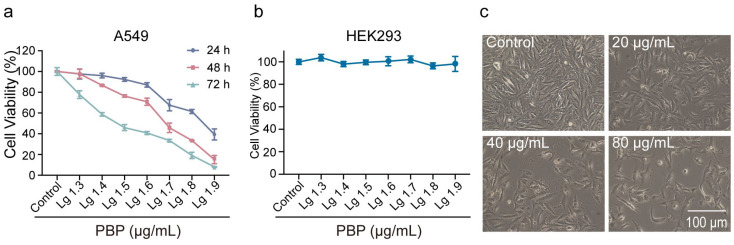
Cytotoxicity of PBP to A549 cells. (**a**) A549 cells viability was assayed using the CCK-8 assay, after being exposed to varying amounts of PBP for 24, 48, and 72 h; (**b**) Cytotoxicity of PBP to HEK293 cell in 48 h determined by CCK-8 assays; (**c**) Typical morphological changes in A549 cells after treatment with PBP for 48 h.

**Figure 6 ijms-23-16141-f006:**
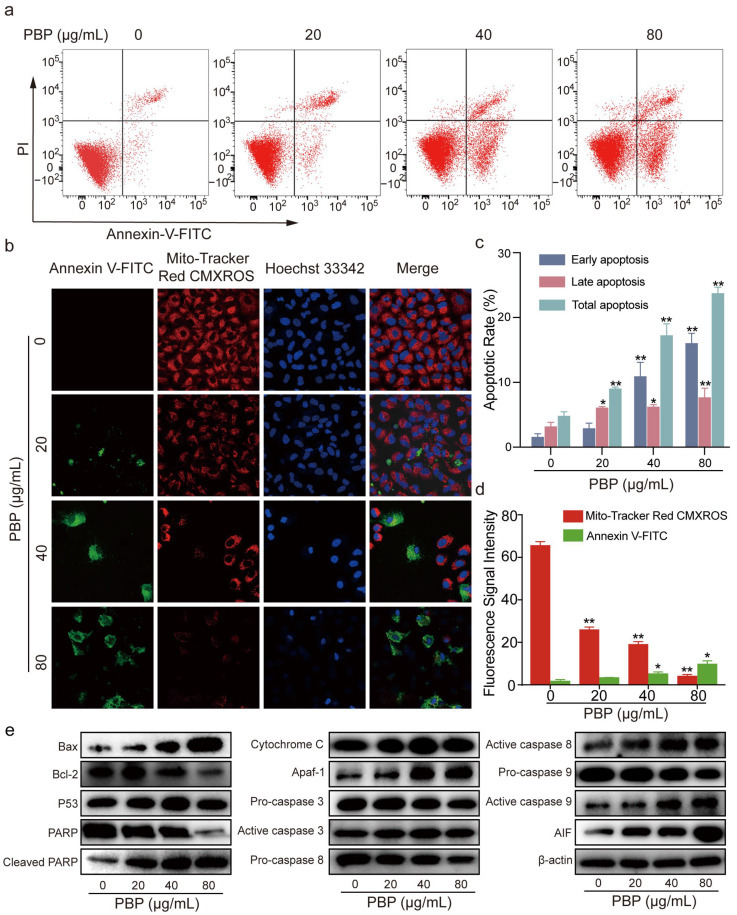
PBP induced apoptosis and caused MMP depolarization in A549 cells. (**a**) A549 cells were incubated with PBP (0, 20, 40 and 80 μg/mL) for 48 h and then stained with Annexin V-FITC and PI for evaluation by flow cytometry; (**b**) The MitoTracker Red CMXRos fluorescent probe was used to detect the MMP in A549 cells using a mitochondrial membrane potential detection kit and laser confocal microscopy; (**c**) Histograms of the apoptotic rates of A549 cells treated with PBP; (**d**) Histograms were used to determine the average fluorescence intensity; (**e**) Expression of apoptosis-related proteins was analyzed by western blot analysis. Data are presented as the mean ± SD (*n* = 3) by One-way ANOVA analyses. ** *p* < 0.01, * *p* < 0.05 versus control (0 μg/mL PBP).

**Figure 7 ijms-23-16141-f007:**
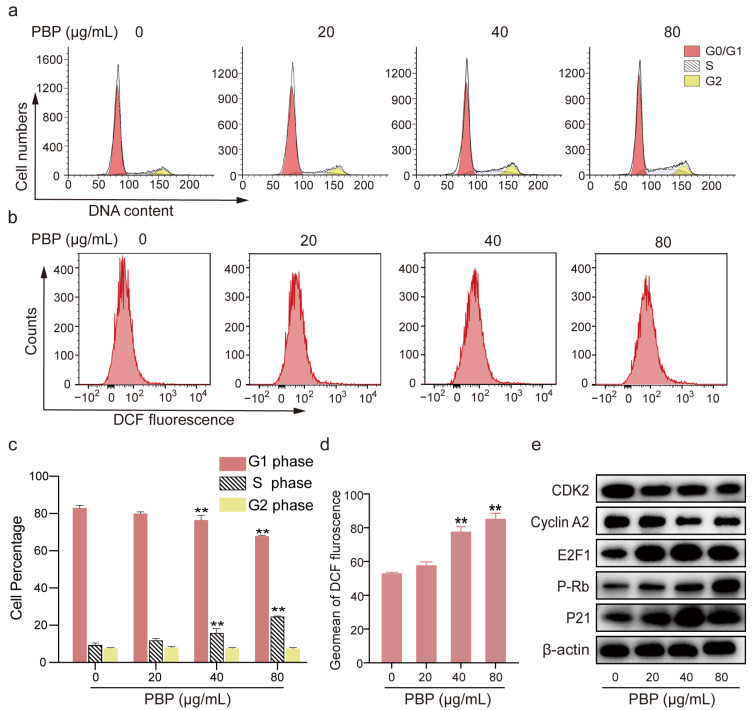
PBP arrested A549 cells in S phase and caused ROS production. (**a**) A549 cells were stained with PI and evaluated using flow cytometry after PBP (0, 20, 40 or 80 μg/mL) treatment for 48 h; (**b**) A549 cells were stained with the ROS indicator (DCF-DA) and analyzed by flow cytometry after PBP treatment for 48 h; (**c**) Analysis quantitative of the cell cycle distribution of A549 cells treated with PBP; (**d**) Quantitative analysis of the DCF fluorescence geomean in A549 cells treated with PBP; (**e**) Western blot analysis of proteins related to S phase arrest. Data are presented as the mean ± SD (*n* = 3) by One-way ANOVA analyses. ** *p* < 0.01 versus control (0 μg/mL PBP).

**Figure 8 ijms-23-16141-f008:**
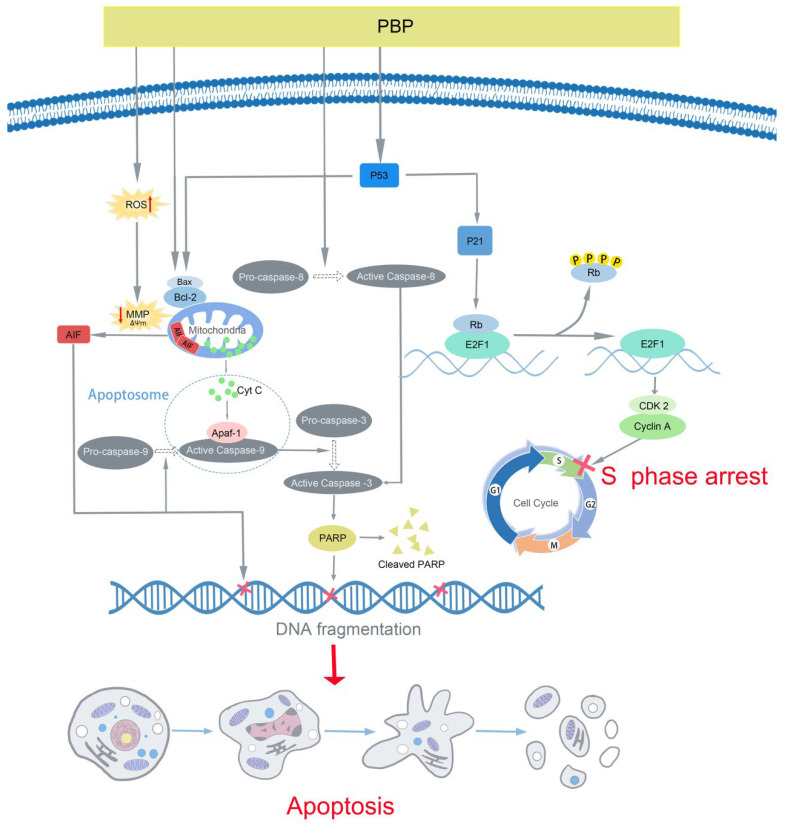
The potential mechanism of PBP exerted antitumor effect by inducing caspase-dependent mitochondrial intrinsic apoptotic pathway and arrested the cell cycle.

**Table 1 ijms-23-16141-t001:** Characterization of the chemical constituents of PBP by UPLC–ESI–QTOF–MS.

Peak No.	Formula	Identification	2D Structure	Peak No.	Formula	Identification	2D Structure
1	C_7_H_6_O_3_	protocatechuic aldehyde	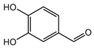	10	C_52_H_32_O_20_	phelligridimer A	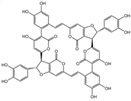
2	C_9_H_8_O_4_	caffeic acid	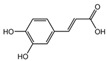	11	C_10_H_10_O_2_	(E)-4-(4-hydroxyphenyl)-3-buten-2-one	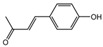
3	C_24_H_20_O_8_	kielcorin	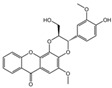	12	C_25_H_20_O_9_	davallialactone	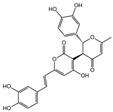
4	C_22_H_16_O_9_	phellibaumin B	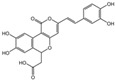	13	C_13_H_14_O_5_	citrinin	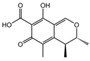
5	C_10_H_10_O_3_	osmundacetone	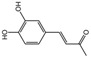	14	C_24_H_20_O_9_	phellibaumin E	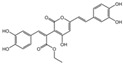
6	C_26_H_18_O_10_	hypholomine B	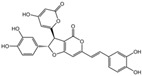	15	C_21_H_14_O_9_	3-(4,6-Dihydroxy-2-oxochromen-3-yl)-8-hydroxy-2-methoxy-2,3-dihydrofuro [3,2-c] chromen-4-one	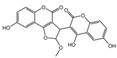
7	C_13_H_10_O_5_	hispidin	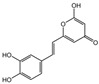	16	C_33_H_20_O_13_	phelligridin I	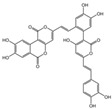
8	C_23_H_18_O_8_	interfungin B	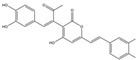	17	C_25_H_18_O_9_	inoscavin A	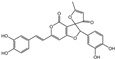

**Table 2 ijms-23-16141-t002:** Degree of active components and binding energy of docking with CASP3, PARP, TP53.

Component	Degree	Binding Energy (CASP3)	Binding Energy (PARP)	Binding Energy (TP53)
protocatechuic aldehyde	26	- ^1^	-	-
caffeic acid	25	-	-	-
osmundacetone	22	−6.4 kcal/mol	−7.2 kcal/mol	−5.3 kcal/mol
hispidin	20	−6.8 kcal/mol	−7.1 kcal/mol	−6.2 kcal/mol
citrinin	15	−6.3 kcal/mol	−6.4 kcal/mol	−6.4 kcal/mol
davallialactone	11	−8.4 kcal/mol	−9.4 kcal/mol	−7.1 kcal/mol

^1^ Molecular docking between the two failed.

**Table 3 ijms-23-16141-t003:** Inhibitory effects of PBP on viability of various tumor cells.

Tumor Cell Lines	A549	HepG2	T24	Hela	HCT116
IC_50_ (μg/mL)	49.1 ± 0.5	140.1 ± 9.6	105.0 ± 10.7	96.1 ± 4.9	54.2 ± 0.7

## Data Availability

Not applicable.

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
