# Peer review of "Phellinus baumii Polyphenol: A Potential Therapeutic Candidate against Lung Cancer Cells"

_ijms, 2022, doi:10.3390/ijms232416141_

Round 1

Reviewer 1 Report

Reviewer comments:

The manuscript provides interesting data regarding the action of polyphenols extracted from Phellinus baumii on lung cancer cells. The manuscript is clear, most relevant for the field because integrated the network pharmacology with experimental verification. The bibliography is in accordance with the research theme addressed by the authors. The research design is well presented and can be easily reproduced. I recommend the manuscript for publication

Congratulations for the amount of work and the approach to the topic!

Author Response

Dear reviewer,

Thank you very much for your recognition. Your encouraging comments have bolstered my confidence and enthusiasm for scientific inquiry.

Kind regards,

Liangen Shi

Reviewer 2 Report

The authors did a thorough study on the characterization and in vitro evaluation of Phellinus baumii on lung cancer cells. This article discussed the identification of the components of PBP, gene functions/molecular interaction, and the evaluation of PBP’s effect on cell apoptosis and corresponding cellular pathways. Overall, the manuscript did a comprehensive investigation regarding the fundamental mechanism of PBP on lung cancer cells and will attract wide interest in this field. Here are some suggestions for the authors.

1.       The authors did a great job on the investigation of apoptosis pathways. As sections 2.9 and 2.11 both discussed the mitochondria-dependent pathway in the mechanism of action of PBP, I recommend the authors reorganize the orders of corresponding sections.

2.       To help readers understand the results better, I suggest the authors to further illustrate the results. For example, the authors should identify early apoptosis and late apoptosis in the flow cytometry results as Annexin V+ PI- and Annexin V+PI+. Also, the methods used for the statistics need to be addressed (student t-test or ANOVA, etc).

3.       Do the authors evaluate PBP on other cancer cell lines or is there literature investigating the cytotoxicity profile of PBP on different cancer types?

4.       Do you have any preliminary in vivo results of PBP? If not, could the author include more discussion regarding in vivo performance/potential off-target cytotoxicity of PBP? 

Author Response

Dear reviewer,

Thank you very much for your professional comments and thoughtful suggestions on our manuscript, these comments are very helpful to improve the quality of the manuscript. We have studied the comments carefully and carefully revised our manuscript, further clarify the logic of writing for improving the quality of the manuscript. A highlighted revised manuscript has been submitted with the revision position change marked in gray, the words in blue are the changes and additions. You can check it in the attached file “Manuscript-Revision”.  I response the comments with a point by point and highlight the changes in revised manuscript in the attached file “Response to Reviewer 2 Comments- MDPI”. Full details of the files are listed. 

Thank you again for pointing out the areas in which our article needs improvement. These suggestions not only help us improve the quality of this manuscripts, but also inspire some new ideas for experimental design and article writing. We sincerely hope that you find our responses and modifications satisfactory. We would be glad to revise the manuscript further, if necessary.
